# Associations of Passive Drinking with Perceived Health Status, Mental Health, and Family Wellbeing in Hong Kong Chinese Adolescents: A Cross-Sectional Study

**Siu Long Chau** [1] , **Yongda Wu** [1] , **Man Ping Wang** [1,*] **and Sai Yin Ho** [2]

1   School of Nursing, University of Hong Kong, Hong Kong, China
2   School of Public Health, University of Hong Kong, Hong Kong, China
*   Correspondence: mpwang@hku.hk; Tel.: +852-3917-6636

**Abstract:** Passive drinking is prevalent in adolescents worldwide, but its prevalence and harm are understudied. Secondary students (*n* = 5840, grades 7–12) from 23 selected schools in Hong Kong participated in the survey from 2015–16. Students reported the harm of passive drinking, perceived health status (Patient Health Questionnaire-2 and Perceived Stress Scale-4), perceived happiness, and family health, happiness, and harmony in the questionnaire. The associations were analyzed using multivariable logistic regression (odds ratio, OR) and linear regression (unstandardized coefficient, b), adjusted for confounders. It was found that 29.1% (95% CI 27.8 to 30.5%) of students experienced passive drinking in the past 30 days. The past 30-day parental passive drinking was associated with a higher level of depressive symptoms (AOR 1.63, 95% CI 1.26 to 2.10), stress (adjusted b 0.76, 0.42 to 1.10), and lower level of perceived happiness (adjusted b −0.52, −0.72 to −0.33). The past 30-day parental passive drinking was associated with a lower level of family health (adjusted b −1.39, 95% CI −1.66 to −1.11), family happiness (adjusted b −1.36, −1.64 to −1.08), and family harmony (adjusted b −1.40, −1.70 to −1.10). Passive drinking was associated with poorer mental health, family wellbeing, and a lower level of happiness among Hong Kong Chinese adolescents.

**Keywords:** passive drinking; perceived health status; mental health; family wellbeing; adolescents; Hong Kong Chinese

## 1. Introduction

Adolescence is one of the most critical periods in life, and it is vulnerable to harmful health behaviors, including excessive alcohol use, smoking, and illicit substance use [1]. Alcohol misuse affects not only the drinkers themselves but also the people around them. Passive drinking (PD) is defined as the impacts of alcohol on people around the drinkers, for example, sleep interruption, child neglect, and verbal abuse [2,3]. It can also contribute to serious harm to the victims, including physical violence, sexual assaults, and unwanted sexual intercourse [3]. In a representative sample of 17,154 Australian adolescents, the most common harm of PD was psychological harm (39%), followed by physical neglect (38%), physical assaults (27%), and sexual abuse (12%) [2]. A study found that girls were at greater risk of experiencing harm involving harassment (e.g., sexual harassment), and boys were more likely to experience harm involving aggression (e.g., physical abuse) from others' drinking [2]. Evidence suggested that adolescents were the most vulnerable group affected by those harms, and the prevalence of PD has increased globally from 12% to 16% in a decade [4–6]. In 2012, approximately 11,000 New Zealand children's hospitalizations were caused by alcohol-related injuries, and their main causes of alcohol-attributable death were road traffic accidents and child abuse by drinking parents [7,8]. Adolescents also had the highest rate of per capita alcohol-related emergency department presentations compared with other age groups [7]. The World Health Organization (WHO) estimated that around

one million adolescents are adversely affected by PD annually worldwide, and the harmful effects persist in adulthood if no intervention is taken [5].

Several studies examined the effects of PD on children's development and wellbeing. Studies found that parental drinking was strongly associated with children's mental health problems, including depression and anxiety [9]. In Hong Kong (HK), a local study showed that exposure to parental pro-drinking practices (e.g., asked by parents to open alcohol bottles) was common in Chinese adolescents [10]. Of 1700 HK secondary students, 51% of them reported seeing their parents drink at home, and 23% of them were asked by parents to open alcohol bottles [10]. Children with drinking parents were more likely to have psychological distress (e.g., stress, anxiety, and depression) because of the higher risk of suffering from verbal abuse and physical violence at home [11]. A study found that one in ten children with drinking parents felt ashamed by their parent's drunken behaviors and felt unsafe when they witnessed their drinking parents having family conflicts at home [11]. Children with drinking parents suffered from societal stigmatization, worsening their psychological distresses [12]. These distresses contributed to various behavioral problems in adolescents, including social withdrawal, aggressive behaviors, and underperforming at school [12]. In addition, verbal or physical abuse by drinking parents caused family conflicts and disrupted family harmony at home, which further affected children's mental and physical health development [13–16].

Since the HK government implemented zero tax on beer and wine in 2008 to boost the alcohol trade, drinking was increasingly promoted and became socially acceptable [10]. Parental pro-drinking practices were common, but the harms of PD in adolescents were not known. Most of these studies were based in European and North American countries, and information on PD in Asian regions was scarce. Traditional alcohol research focused on alcohol's harmful effects on the drinkers themselves, and the drinking harms to others was understudied. Only one study was found in Pubmed and the Cochrane Database of Systematic Reviews examining the harms of PD on adolescents in Asian regions [5]. Using a large representative sample of HK Chinese adolescents, the present study aims to investigate the overall prevalence of PD, and identify the associations of PD with perceived health status, mental health, perceived happiness, and family wellbeing in this population.

## 2. Methods

### 2.1. Study Design

In this cross-sectional study, we randomly sampled one local school from every 18 districts and one international school from every five regions (Hong Kong Island, Kowloon East, Kowloon West, New Territories East, and New Territories West) using a sampling frame provided by the Education Bureau covering all schools in HK. Twenty-three schools (of the 349 invited, 6.6%) participated; refusal was mainly due to time and administrative issues. Two classes from each of the six grades (Grade 7–12) were randomly selected in the participating schools. Invitation letters were sent to parents to explain the purpose of the survey. Parents and students who refused to participate were asked to return the blank questionnaire during the survey. Students' participation was voluntary, even with parental consent. Their refusal rate was not estimated, as most of them did not return the blank questionnaires on the day of the survey. A self-administered anonymous questionnaire was used to collect data from students, which took 20 to 30 min to complete. Teachers were reminded not to influence students' answers in the classroom. The completed questionnaire was inserted into a small opaque envelope; all envelopes were put into an opaque bag and sealed immediately after completion. Postage-paid envelopes containing a blank questionnaire were prepared for absentees on the day of the survey; they were asked to return the completed questionnaire to the research team directly by post. Students who completed the questionnaire ($n = 5840$) were analyzed. Ethical approval was granted by the Institutional Review Board of the University of Hong Kong/Hospital Authority Hong Kong West Cluster (UW 14-509).

*2.2. Measurements*

Socio-demographic characteristics (sex, age, place of birth, and perceived family affluence) and drinking behaviors (age of first drinking, drinking frequency, drinking quantity, and binge drinking frequency) were collected in the questionnaire. Students also reported the harm of passive drinking, perceived health status, mental health (Patient Health Questionnaire-2 and Perceived Stress Scale-4), perceived happiness, and family wellbeing in the questionnaire.

Current passive drinking was defined as experiencing the harm of passive drinking in the past 30 days, and ever passive drinking was defined as experiencing its harm in one's lifetime. Passive drinking was assessed by 16 items: noise, study/sleep interruption, felt troubled by littering, exposure to vomit or urination, felt neglected, emotionally hurt, felt unsafe, took care of a drunk person, verbal insult or harassment, physical assault, sexual harassment, unwanted intercourse, properties damaged, accidents, financial loss, and others. We also collected the causes of ever and past 30-day passive drinking of students in the questionnaire, including the passive drinking caused by parents, siblings, relatives, peers, and others.

Physical health was assessed by self-perceived health status. "Very well" and "well" responses were categorized as good health. "Fair", "bad", and "very bad" responses were categorized as poor or fair health. Mental health was assessed by Patient Health Questionnaire-2 (PHQ-2) and Perceived Stress Scale-4 (PSS-4). PHQ-2 score of 3 or higher was categorized as more likely to have depressive symptoms [17]. The total score of PSS-4 ranged from 0 to 16; a higher score indicated a higher stress level [18]. Both scales were reliable in measuring depressive symptoms and stress levels in the general population in HK, with Cronbach's alpha of 0.76 and 0.82, respectively [19,20].

Perceived happiness was assessed by the Subjective Happiness Scale (SHS) with a total score ranging from 1 to 7; a higher score indicated a higher happiness level [21]. The scale showed high reliability in measuring happiness in the general population in HK, with Cronbach's alpha of 0.82 [21]. Family wellbeing is a scale combined with scales of family health, family happiness, and family harmony. Each score ranged from 0 to 10; higher scores indicated better family wellbeing [22]. The scale was reliable with Cronbach's alpha of 0.84 [22].

*2.3. Statistical Analysis*

A total of 283 students (4.6%) with missing answers for over half of the questionnaire were excluded. Students who provided inconsistent answers were also excluded, leaving 5840 students for analysis. All descriptive data and prevalence of passive drinking were weighted by sex and age distribution of students in HK based on EDB 2014–15 student enrollment statistics [23]. The associations of ever and past 30-day passive drinking with perceived health status, PHQ-2, PSS-4, SHS, and family wellbeing were analyzed using multivariable logistic regression (adjusted odds ratio, AOR) and multiple linear regression (adjusted unstandardized coefficient, b), controlling for sex, age, perceived family affluence, and current drinking status. Stata 15.1 was used for all analyses. A two-tailed *p*-value less than 5% was considered statistically significant.

## 3. Results

Table 1 shows that 51.5% of participants were male, 64.0% were aged less than 16 years, 69.9% were born in HK, and 55.8% perceived their family affluence as average; 46.5% engaged in underage drinking, 23.5% drank at least monthly, 14.4% drank more than 1 unit of alcohol on a drinking day, and 17.7% engaged in binge drinking.

**Table 1.** Socio-demographic characteristics and drinking behaviors of 5840 participants [a].

|  | *n* [b] | % (95% CI) |
|---|---|---|
| Sex |  |  |
| Male | 3239 | 51.5 (50.0, 53.0) |
| Female | 2601 | 48.5 (47.0, 50.0) |
| Age |  |  |
| 11–15 | 3513 | 64.0 (62.6, 65.3) |
| 16–20 | 2327 | 36.0 (34.7, 37.4) |
| Place of birth |  |  |
| Hong Kong | 4002 | 69.9 (68.5, 71.2) |
| Mainland China | 1397 | 22.4 (21.3, 23.6) |
| Others | 408 | 7.7 (6.8, 8.7) |
| Perceived family affluence |  |  |
| Below average | 1473 | 25.4 (24.1, 26.7) |
| Average | 3153 | 55.8 (54.3, 57.3) |
| Above average | 1048 | 18.8 (17.6, 20.0) |
| Age of first drinking 1 unit of alcohol [c] |  |  |
| Never | 2741 | 53.4 (51.9, 54.9) |
| 7–11 | 1529 | 25.9 (24.7, 27.2) |
| 12 or above | 1092 | 20.6 (19.4, 21.9) |
| Drinking frequency |  |  |
| Never | 2972 | 53.2 (51.7, 54.6) |
| Yearly or less | 1388 | 23.3 (22.1, 24.6) |
| Monthly or less | 780 | 12.4 (11.5, 13.2) |
| 1–3 times per month | 529 | 8.3 (7.7, 9.1) |
| 1–6 times per week | 110 | 1.7 (1.4, 2.1) |
| Everyday | 47 | 1.1 (0.6, 1.7) |
| Usual drinking quantity in a drinking day [c] |  |  |
| 0 | 2777 | 53.5 (52.0, 55.0) |
| Less than 0.5 unit | 809 | 14.6 (13.7, 15.7) |
| 0.5–1 unit | 998 | 17.4 (16.3, 18.5) |
| 2–4 units | 556 | 9.5 (8.7, 10.4) |
| 5 or more units | 278 | 4.9 (4.3, 5.7) |
| Drinking ≥ 5 units of alcohol on one occasion [c] |  |  |
| Never | 4752 | 82.2 (81.1, 83.4) |
| Less than yearly | 612 | 10.3 (9.4, 11.2) |
| Yearly | 262 | 4.0 (3.6, 4.6) |
| Monthly | 143 | 2.4 (2.0, 2.9) |
| Weekly | 27 | 0.5 (0.3, 1.0) |
| Daily | 25 | 0.5 (0.3, 1.0) |

[a] Prevalence was weighted by the sex and age distribution of students in Hong Kong. [b] Observation (*n*) is not 5840 due to nonresponse. [c] 1 unit equals 10 mL of pure alcohol.

Table 2 shows the overall prevalence of ever (40.8%, 95% CI 39.4 to 42.2%) and past 30-day (29.1%, 27.8 to 30.5%) PD. The most prevalent past 30-day PD was noise (14.1%, 95% CI 13.0 to 15.2%); 12.8% (11.9 to 13.8%) felt troubled by littering, 9.5% (8.7 to 10.4%) felt neglected, and 9.3% (8.5 to 10.3%) experienced verbal insult or harassment; 1.4% (1.1 to 1.9%) experienced sexual harassment, and 0.6% (0.4 to 0.8%) experienced unwanted intercourse in the past 30 days.

Table 3 shows that ever—siblings PD and ever—peer PD were significantly associated with a lower level of perceived happiness in general. Ever—relative PD was significantly associated with higher odds of depressive symptoms and a lower level of perceived happiness compared with peers. PD in the past 30 days was significantly associated with higher odds of depressive symptoms, a higher stress level, a lower level of perceived happiness in general, and a lower level of perceived happiness compared with peers. Increased exposures to PD caused by parents, siblings, relatives, peers, and others were associated with increasing odds of having depressive symptoms, and a decreasing level of perceived happiness in general (*p* trend < 0.05).

**Table 2.** Prevalence of passive drinking of 5840 participants [a].

| Passive Drinking | Ever | | Past 30-Day | |
|---|---|---|---|---|
| | *n* | % (95% CI) | *n* | % (95% CI) |
| Overall | 2397 | 40.8 (39.4, 42.2) | 1691 | 29.1 (27.8, 30.5) |
| Noise | 1338 | 23.2 (22.0, 24.5) | 795 | 14.1 (13.0, 15.2) |
| Study/sleep interrupted | 1112 | 19.2 (18.1, 20.4) | 751 | 12.8 (11.9, 13.8) |
| Felt troubled by littering | 545 | 9.2 (8.4, 10.0) | 319 | 5.4 (4.8, 6.1) |
| Exposed to vomit or urination | 572 | 9.7 (8.9, 10.6) | 224 | 3.9 (3.3, 4.5) |
| Felt neglected | 818 | 14.2 (13.2, 15.3) | 548 | 9.5 (8.7, 10.4) |
| Emotionally hurt | 808 | 13.5 (12.6, 14.4) | 488 | 8.2 (7.5, 9.0) |
| Felt unsafe | 741 | 12.8 (11.9, 13.8) | 408 | 7.1 (6.4, 7.8) |
| Took care of a drunk person | 705 | 11.9 (11.0, 12.9) | 266 | 4.6 (4.0, 5.3) |
| Verbal insult or harassment | 851 | 15.0 (13.9, 16.1) | 529 | 9.3 (8.5, 10.3) |
| Pushed, hit, or assaulted | 464 | 8.5 (7.7, 9.5) | 268 | 5.0 (4.3, 5.8) |
| Sexual harassment | 167 | 3.0 (2.5, 3.6) | 80 | 1.4 (1.1, 1.9) |
| Unwanted intercourse | 83 | 1.5 (1.1, 1.9) | 38 | 0.6 (0.4, 0.8) |
| Properties damaged | 266 | 4.9 (4.2, 5.7) | 141 | 2.8 (2.2, 3.5) |
| Accidents | 110 | 2.0 (1.6, 2.6) | 43 | 0.8 (0.5, 1.2) |
| Financial loss | 232 | 3.9 (3.4, 4.5) | 127 | 2.1 (1.7, 2.5) |
| Others | 149 | 2.6 (2.1, 3.2) | 65 | 1.2 (0.9, 1.7) |

[a] Prevalence was weighted by the sex and age distribution of students in Hong Kong.

**Table 3.** Associations of causes of ever and past 30-day passive drinking with perceived health status, mental health status, and perceived happiness in 5840 participants.

| Causes of Ever/Past 30-Day PD [a] | Perceived Health Status [b] (Fair or Poor) | Patient Health Questionnaire-2 [c] ($\geq$3) | Perceived Stress Scale-4 [d] | Perceived Happiness in General [e] | Perceived Happiness Compared with Peers [e] |
|---|---|---|---|---|---|
| | Adjusted OR (95% C.I.) [f] | | | Adjusted b (95% C.I.) [f] | |
| **Parental PD** | | | | | |
| No PD (*n* = 5455) | 1 | 1 | 0 | 0 | 0 |
| Parental ever PD (*n* = 113) | 1.09 (0.70, 1.69) | 0.90 (0.59, 1.39) | 0.52 (−0.001, 1.04) | −0.11 (−0.41, 0.19) | −0.24 (−0.54, 0.07) |
| Parental PD in past 30 days (*n* = 272) | 1.26 (0.96, 1.66) | 1.63 (1.26, 2.10) *** | 0.76 (0.42, 1.10) *** | −0.52 (−0.72, −0.33) *** | −0.37 (−0.57, −0.17) *** |
| *p* trend | 0.09 | <0.01 | <0.001 | <0.001 | <0.001 |
| **Siblings PD** | | | | | |
| No PD (*n* = 5665) | 1 | 1 | 0 | 0 | 0 |
| Siblings ever PD (*n* = 21) | 1.28 (0.47, 3.49) | 0.57 (0.19, 1.71) | −0.04 (−1.21, 1.13) | −0.78 (−1.45, −0.10) * | −0.60 (−1.29, 0.08) |
| Siblings PD in past 30 days (*n* = 154) | 0.90 (0.62, 1.33) | 1.50 (1.07, 2.11) * | 0.08 (−0.36, 0.52) | −0.30 (−0.56, −0.04) * | −0.17 (−0.43, 0.09) |
| *p* trend | 0.67 | <0.05 | 0.74 | <0.01 | 0.12 |
| **Relatives PD** | | | | | |
| No PD (*n* = 5717) | 1 | 1 | 0 | 0 | 0 |
| Relatives ever PD (*n* = 40) | 1.57 (0.80, 3.09) | 2.18 (1.15, 4.12) * | 0.74 (−0.11, 1.60) | −0.48 (−0.97, 0.02) | −0.66 (−1.16, −0.15) * |
| Relatives PD in past 30 days (*n* = 83) | 1.07 (0.64, 1.77) | 1.94 (1.23, 3.04) ** | 0.25 (−0.35, 0.86) | −0.73 (−1.07, −0.38) *** | −0.40 (−0.76, −0.05) * |
| *p* trend | 0.51 | <0.01 | 0.18 | <0.001 | <0.01 |
| **Peers PD** | | | | | |
| No PD (*n* = 5181) | 1 | 1 | 0 | 0 | 0 |
| Peers ever PD (*n* = 114) | 0.97 (0.63, 1.50) | 0.99 (0.65, 1.50) | 0.37 (−0.14, 0.89) | −0.34 (−0.64, −0.05) * | −0.16 (−0.47, 0.14) |
| Peers PD in past 30 days (*n* = 545) | 0.83 (0.67, 1.03) | 1.40 (1.16, 1.70) *** | 0.30 (0.05, 0.54) * | −0.27 (−0.41, −0.12) *** | −0.24 (−0.39, −0.10) ** |
| *p* trend | 0.09 | <0.01 | <0.05 | <0.001 | <0.01 |
| **Others PD** | | | | | |
| No PD (*n* = 4962) | 1 | 1 | 0 | 0 | 0 |
| Others ever PD (*n* = 176) | 0.83 (0.57, 1.21) | 1.11 (0.79, 1.56) | 0.17 (−0.25, 0.59) | 0.05 (−0.19, 0.30) | 0.002 (−0.25, 0.25) |
| Others PD in past 30 days (*n* = 702) | 1.12 (0.93, 1.34) | 1.43 (1.20, 1.69) *** | 0.35 (0.13, 0.57) ** | −0.19 (−0.32, −0.06) ** | −0.09 (−0.22, 0.03) |
| *p* trend | 0.33 | <0.001 | <0.01 | <0.01 | 0.17 |

[a] PD stands for passive drinking. Participants were mutually exclusive. [b] Reference group: Perceived health status rated as good, very good, and excellent. [c] Reference group: Score 0–2. A score $\geq$ 3 suggests the participants may have depression. [d] Total score: 0–16. A higher score suggests more perceived stress. [e] Total score: 1–7. A higher score suggests greater perceived happiness. [f] Adjusted for sex, age, perceived family affluence, and current drinking status. * $p < 0.05$; ** $p < 0.01$; *** $p < 0.001$.

Table 4 shows that ever—parental PD was significantly associated with lower family health, family happiness, family harmony, and poorer family wellbeing. Ever—siblings PD was significantly associated with lower family health, family happiness, and poorer family wellbeing. Past 30-day parental PD was significantly associated with lower family

health, family happiness, family harmony, and poorer family wellbeing. Increased exposure to parental PD was associated with decreasing levels of family health, family happiness, family harmony, and family wellbeing (*p* trend < 0.001).

**Table 4.** Associations of ever and past 30-day family-caused passive drinking with family health, happiness, harmony, and wellbeing of 5840 participants.

| Causes of Ever/Past 30-Day PD [a] | Family Health [b] | Family Happiness [b] | Family Harmony [b] | Family Wellbeing [c] |
|---|---|---|---|---|
| | Adjusted b (95% C.I.) [d] | | | |
| Parental PD | | | | |
| No PD (*n* = 5455) | 0 | 0 | 0 | 0 |
| Parental ever PD (*n* = 113) | −0.56 (−0.98, −0.14) ** | −0.52 (−0.95, −0.08) ** | −0.57 (−1.03, −0.12) ** | −1.66 (−2.90, −0.42) * |
| Parental PD in past 30 days (*n* = 272) | −1.39 (−1.66, −1.11) *** | −1.36 (−1.64, −1.08) *** | −1.40 (−1.70, −1.10) *** | −4.19 (−5.00, −3.38) *** |
| *p* trend | <0.001 | <0.001 | <0.001 | <0.001 |
| Siblings PD | | | | |
| No PD (*n* = 5665) | 0 | 0 | 0 | 0 |
| Siblings ever PD (*n* = 21) | −1.34 (−2.30, −0.39) ** | −1.23 (−2.21, −0.24) * | −1.00 (−2.03, 0.03) | −3.58 (−6.39, −0.77) * |
| Siblings PD in past 30 days (*n* = 154) | −0.18 (−0.54, 0.18) | −0.31 (−0.68, 0.07) | −0.28 (−0.67, 0.12) | −0.75 (−1.83, 0.32) |
| *p* trend | 0.14 | <0.05 | 0.09 | 0.07 |

[a] PD stands for passive drinking. Participants were mutually exclusive. [b] Total score: 0–10. A higher score suggests more perceived health, happiness, and harmony. [c] Combined scores of family health, happiness, and harmony. Total score: 0–30. A higher score indicated better family wellbeing. [d] Adjusted for sex, age, and current drinking status. * *p* < 0.05; ** *p* < 0.01; *** *p* < 0.001.

## 4. Discussion

This is the first large-scale study to examine the prevalence of PD and its association with perceived health status, mental health, perceived happiness, and family wellbeing from a large representative sample of HK Chinese adolescents. We observed that the prevalence of current PD (29.1%) was lower than in Australia (33%) but was higher than in some Pacific and Asian countries, such as New Zealand (22%) and Vietnam (13.9%) [5]. The non-bodily harm of PD was more common than the bodily harm, and the results were consistent with studies from western countries (e.g., Finland, Ireland, and Scotland) [24]. The most common harm was psychological harm (e.g., verbal abuse), followed by severe harm, including physical violence and sexual assaults [2,24]. Our study found that about one-third of adolescents in HK were affected by the harm of PD, and interventions are warranted to educate adolescents to avoid those harms.

Some studies examined the associations of PD with adolescent health development, including behavioral and mental health problems [25,26]. Consistent with previous studies [25,26], our findings showed that adolescents who experienced current PD had higher odds of depressive symptoms, stress, and lower happiness levels. Studies suggested that adolescents' development was shaped by the external environment, and their mental health was strongly affected by others' drinking, especially by peers and family members with alcohol misuse problems [27]. Adolescents with peers or parents diagnosed with alcohol misuse problems had a higher risk of developing mood disorders due to the experience of different types of abuse (e.g., domestic violence, sexual assaults, and bullying) [25]. These traumatizing events were associated with poor mental health [25]. Our study also contributed to previous research by showing that the associations of PD with mental health problems and happiness may be dose-dependent and we extended the understanding of the different causes of PD as a determinant of adolescents' mental health problems and lower happiness. The robust findings suggested that PD was a significant predictor of poorer mental health development and lower happiness in Chinese adolescents.

We extend the understanding of PD to family wellbeing. We found that adolescents who experienced PD from parents and siblings were associated with lower family health, happiness, and harmony. Pouring alcohol for seniors and drinking in front of children at home were considered a social norm in Chinese culture and adopted by most HK parents [26]. Nevertheless, studies found that drinking parents were more likely to engage

in verbal aggression and have unresolved family conflicts at home [28]. Parents drinking in front of their children at home had a higher risk of physical assault and sexual abuse on family members compared with non-drinking parents and was associated with poorer family relations [13,14]. An intact family environment with role models is essential to children's normal physical and mental health development [14]. Parents should be warned that drinking may contribute to family disharmony and unhappiness, and encouraged to seek help from family counselling services and not to resort to alcohol to manage unresolved family conflicts. We found that the associations of parental PD and family wellbeing were dose-dependent; it suggested that parental PD had a significant influence on family relationships. Our robust findings showed that family-based interventions are needed to educate parents on the harm of PD to their children and family harmony.

This study has some limitations. First, the data collected were self-reported, but we encouraged candid reporting by providing an opaque envelope to seal the completed questionnaires. Second, the representativeness of the study may be affected as not all forms in some schools participated in the survey and the refusal rate was high, but we calculated the sex- and age-weighted prevalence based on the years' enrollment statistics and used a random sampling method to recruit schools from all 18 districts and all five regions of HK to increase the representativeness of the sample and minimize the systematic sampling bias. Third, the analysis was based on cross-sectional data, and reverse causality could not be ruled out. A longitudinal study is needed to further explore the temporal relations in the observed associations.

## 5. Conclusions

Our study found that one-third of HK Chinese adolescents experienced PD in the past 30 days, and some even experienced physical assaults, sexual harassment, and unwanted sexual intercourse from others' drinking. Adolescents who currently experienced PD from parents and peers were associated with a higher level of depressive symptoms, stress, and a lower level of perceived happiness. PD caused by parents and siblings was also associated with poorer family wellbeing. We found that PD was common among HK Chinese adolescents, and a variety of drinking-related behaviors, including parents drinking in front of children, were associated with the harms of PD in students. Our study provided implications that school- and family-based interventions are needed to educate parents to avoid drinking at home and help stop relatives from offering alcohol to their children. School counseling should screen for students' psychological distress caused by PD to promote healthy coping strategies. Our findings also informed policies to reduce the harms of PD in students, including school anti-alcohol policies (e.g., banning alcohol at school), limiting the number of alcohol outlets, restrictions on the hours of alcohol sale, and a tax increase on alcohol prices to curtail the attractiveness of alcohol and reduce the subsequent harms of PD on adolescents. Future studies may build on the present findings and investigate whether longitudinal associations between PD and its harms exist. An ecological momentary assessment could also be used to measure the exposures of PD more precisely and frequently.

**Author Contributions:** Conceived and designed the study, M.P.W. and S.Y.H.; collected the data, M.P.W., S.Y.H. and Y.W.; analyzed the data, S.L.C. and Y.W.; wrote the first draft of the manuscript, S.L.C. All authors have read and agreed to the published version of the manuscript.

**Funding:** This research received no external funding.

**Institutional Review Board Statement:** This study was conducted in accordance with the Declaration of Helsinki, and approved by the Institutional Review Board of the University of Hong Kong/Hospital Authority Hong Kong West Cluster (UW 14-509).

**Informed Consent Statement:** Informed consent was obtained from all participants involved in the study. Written informed consent has been obtained from the participants to publish this paper.

**Data Availability Statement:** The dataset generated during and/or analyzed during the current study are available from the corresponding author upon reasonable request.

**Acknowledgments:** We thank the students for their participation in this study.

**Conflicts of Interest:** The authors declare no conflict of interest.

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
