# Peer review of "Associations of Passive Drinking with Perceived Health Status, Mental Health, and Family Wellbeing in Hong Kong Chinese Adolescents: A Cross-Sectional Study"

_adolescents, doi:10.3390/adolescents3010013_

Round 1

Reviewer 1 Report

I have carefully reviewed the revised manuscript adolescents-2217467 with Dr. Chau as the first author. The title is " Associations of Passive Drinking with Perceived Health Status, Mental Health, and Family Wellbeing in Hong Kong Chinese Adolescents: A Cross-Sectional Study.

Passive drinking is one of the alcohol-related problems that should be feared, yet very few studies have been reported on it. Therefore, there is no doubt that this study is very significant. The results are clearly presented and the conclusions are hardly controversial.

As noted in the limitations of the study, the survey appears to have been properly researched, although the methodology of the survey may affect the results. We look forward with great anticipation to further research, including longitudinal studies.

Reviewer 2 Report

Conments:

1.     The review of the literature review in the preface of the first part is inadequate, and the basis for the selection of the topic is not sufficient and needs to be added.

2.     The research question of this paper is not presented.

3.     At a minimum, the reliability of the test instrument should be reported in this study.

4.     What is the rationale for using these two questionnaires of Patient Health Questionnaire-2 (PHQ-2) and Perceived Stress Scale-4 (PSS-4) to reflect mental health?

5.     If the three scales of Subjective Happiness Scale (SHS), Patient Health Questionnaire-2 (PHQ-2) and Perceived Stress Scale-4 (PSS-4) can be used in the study, you should report the reliability of these three instruments at least.

Reviewer 3 Report

Thank you for the opportunity to read the above-named manuscript. The manuscript focues on passive drinking in adolescents living in Hong Kong. The manuscript is well-written and contains interesting findings. Nonetheless, I have some comments that may help to strengthen the manuscript:

Abstract: Should not contain headings (according to author guidelines)

·         Introduction: Please use the same number of decimals for all proportions reported.

·         Study design: Refusal rates not only of schools, but also of parents / adolescents would be interesting

·         line 65:  Refusal rate is very high; this might rise concerns for a systematic sampling bias that should be addressed in the limitations.

·         lines 109 f.: Student enrollment statistics should be cited.

·         Measurements: Analyses in Results part are conducted using subgroups of passive drinking (e.g. parental, siblings). It should be included that these subgroups of PD were collected as well.

·         Tables seem to have some formatting issues that should be fixed.

·         lines 133-147 and lines 157-166.: As regression estimates are already presented in tables 3 and 4, reported estimates in these paragraphs might be redundant. Excluding them would increase readability of these paragraphs as well.

·         Tables 3 and 4: A better structure would increase readability (e.g. headings for parental, siblings, relatives PD);

·         Tables 3 and 4: Is there a reason, why p trend is presented in addition to significance stars? If so p trend should be addressed in the Results part.

·         line 224: typo (harms)

·         line 196: language (traumatized events -> traumatic/traumatizing events)

·         Conclusions: What are implications for prevention and future research? Please include this in the conclusions section.
